# The Innate Defense in the Zika-Infected Placenta

**DOI:** 10.3390/pathogens11121410

**Published:** 2022-11-24

**Authors:** Laíza Vianna Arruda, Natália Gedeão Salomão, Felipe de Andrade Vieira Alves, Kíssila Rabelo

**Affiliations:** 1Interdisciplinary Laboratory of Medical Research, Instituto Oswaldo Cruz, Oswaldo Cruz Foundation, Rio de Janeiro 21040900, Brazil; 2Department of Histology and Embryology, Roberto Alcantara Gomes Institute of Biology, Rio de Janeiro State University, Rio de Janeiro 20551030, Brazil

**Keywords:** Zika virus, placenta, immune response, innate immunity

## Abstract

Zika virus (ZIKV) is an arthropod-borne virus that belongs to the *Flaviviridae* family, genus *Flavivirus* and was first isolated 1947 in Uganda, Africa, from the serum of a sentinel Rhesus monkey. Since its discovery, the virus was responsible for major outbreaks in several different countries, being linked to severe complications in pregnant women, neonatal birth defects and the congenital zika syndrome. Maternal–fetal transmission of ZIKV can occur in all trimesters of pregnancy, and the role of the placenta and its cells in these cases is yet to be fully understood. The decidua basalis and chorionic villi, maternal–fetal components of the placenta, contain a rich immunological infiltrate composed by Hofbauer cells, mastocytes, dendritic cells and macrophages, primary cells of the innate immune response that have a role that still needs to be better investigated in ZIKV infection. Recent studies have already described several histopathological features and the susceptibility and permissiveness of placenta cells to infection by the Zika virus. In this review, we address some of the current knowledge on the innate immune responses against ZIKV, especially in the placenta.

## 1. Introduction

Zika fever is an arbovirus transmitted mainly by mosquitoes of the *Aedes* genus (*Stegomyia* subgenus). The zika virus (ZIKV) was discovered in Africa in 1947 from the blood of Rhesus monkeys inhabiting the zika forest [1,2] and was first detected in humans in Asia in 1966, but its potential impact on public health was not recognized until the virus caused outbreaks in the Pacific from 2007 to 2015, the year it began to spread across America [3,4]. In 2019, autochthonous transmission of ZIKV was confirmed in 87 countries or territories in the Americas [4]. Large outbreaks have clearly occurred in many countries and territories due to the introduction of the virus into immunologically virgin populations and with a widespread presence of vectors. Overall, by region, South America represented 70% of reported cases, the Caribbean 21%, Central America 9% and North America 1%. The highest number of suspected and confirmed cases was reported in Brazil (346,475 cases, 46%) followed by Colombia (107,206, 14%) and Venezuela (62,200; 8%) [5]. ZIKV transmission has significantly declined in the Americas since late 2016; fewer than 30,000 cases were reported in 2018, compared with the more than 500,000 cases reported at the height of the pandemic in 2016 [6].

In 2015, Brazil recorded the first outbreak of ZIKV infection, with cases initially reported in the northeast region [7] but rapidly spreading throughout the country. Some studies with retrospective phylogenetic analysis suggest that the introduction of the virus into Brazil may have occurred in 2013 [6,8,9]. Zika fever is a compulsory notification disease and in 2016, 216,207 cases were reported in Brazil, with 8 laboratory-confirmed deaths. More recent data from Brazil’s Ministry of Health show that there has been variation in the numbers of cases and deaths/year, with 17,594 probable cases reported in 2017 with 8 deaths, 8680 cases in 2018 with 5 deaths, and 10,768 cases in 2019 with 3 deaths [10,11]. During this year, zika cases increased in Brazil, with 9916 probable cases until week 32, corresponding to an incidence rate of 4.6 cases per 100,000 inhabitants in the country. Compared with 2019, there was a 21.1% increase in the number of cases and compared to the year 2021, an increase of 98.8% is observed. However, deaths by zika were not reported in the country until the respective week of the year 2022 [10]. This increase in cases among the different arboviruses is seasonal and predicted by epidemiologists, so it is necessary to monitor the number of zika cases and the progress of studies in this area in order for the scientific community to be prepared to deal with outbreaks, epidemics, and their consequences.

Because of the large number of zika cases in Brazil in 2015, it was declared a Public Health Emergency of National Importance [12], and in February 2016, the World Health Organization (WHO) declared it a Public Health Emergency of International Importance [13]. Along with the cases of zika, there have been alarming reports of microcephaly cases associated with the infection [14], which was observed as one of the signs of a new congenital disease resulting from ZIKV infection during pregnancy and was named congenital zika syndrome (CZS). CZS is characterized by a set of congenital, structural and functional anomalies, with repercussions for the growth and development of fetuses exposed to the virus during pregnancy [15]. Because of the many cases of microcephaly and later, the perception of congenital zika syndrome, the studies about this disease intensified. From 2015 to date, 3707 cases of birth defects associated with ZIKV infection have been reported in Brazil and a slightly higher number in America [4,16].

## 2. The Zika Virus

Few studies have reported the evolutionary biology of ZIKV [17,18,19], but such studies describe three main lineages of ZIKV, one from Asia and two from Africa. The etiologic agent of this disease is zika virus, belonging to the genus *Flavivirus*, which is approximately 25–50 nm in size and shares the same family (*Flaviviridae*) as other widely known viruses, such as dengue, West Nile, Japanese encephalitis virus, and yellow fever virus [17,20,21]. Flaviviruses are icosahedral viruses formed by an envelope composed of a lipid bilayer in which the envelope (E) and membrane (M) proteins are inserted. In the inner portion of the viral envelope, there is a nucleocapsid composed of multiple molecules of the capsid protein (C) complexed to the viral genome, a single-stranded RNA molecule with positive polarity [22].

The virus genome is approximately 11 kb in length and has the cap structure at its 5′ end, being devoid of a poly-A tail at the 3′ end, and comprises a single open reading frame encoding a polyprotein precursor to flavivirus proteins. This polyprotein is initially anchored to the endoplasmic reticulum via transmembrane helices that cross the membrane, and is subsequently cleaved by cellular proteases and the viral protease (NS3/2B protein), generating three structural proteins, capsid, premembrane and envelope, and seven nonstructural proteins, NS1, NS2A, NS2B, NS3, NS4A, NS4B and NS5. Proteins C, prM and E are incorporated into the viral particles during their maturation, while the non-structural proteins are involved in the replication and/or assembly of the virions. The 3′ and 5′ non-coding regions (3′UTR and 5′UTR) are also important for viral replication [22,23].

## 3. Zika, Transmission and Clinical Manifestations

In general, most individuals are asymptomatic or develop mild clinical disease [24], which has as classic symptoms headache, fever, myalgia, exanthema, arthralgia and conjunctivitis [4,24,25,26,27].

As with other arboviruses such as dengue (DENV) and chikungunya (CHIKV), ZIKV is transmitted mainly by vectors: *Aedes aegypti* and *Aedes albopictus* [28]. However, transmission of ZIKV can occur through different routes, including sexual transmission, with the presence of virus being detected in semen [29] and in uterine cells [30], evidencing that the virus can be transmitted between both sexes, although women are more susceptible [31].

In addition, the virus has been detected in the blood of infected patients, and its transmission has even been reported in platelet transfusions [32]. Transmission via breast milk can also occur since the presence of ZIKV in breast milk has already been studied and described in some studies, however none of them confirmed transmission [33].

During the 2015 epidemic, the possibility of the vertical transmission of the Zika virus was confirmed, being associated with CZS [34,35,36] Maternal–fetal transmission of ZIKV can occur in all trimesters of pregnancy, whether the infection in the mother is symptomatic or asymptomatic [4]. The way in which ZIKV crosses the placental barrier and infects the fetus has not been fully understood. The mechanisms by which viruses can be transmitted vertically are multifaceted and may involve direct hematogenous, transcellular trophoblastic, or paracellular pathways, in addition to transport within immune cells [37]. The multiplication of Hofbauer cells during placental inflammation caused by the infection is believed to facilitate vertical transmission [38,39]. Some studies have identified the ZIKV genome in the amniotic fluid, intervillar space, decidual and chorionic villi of the placenta, in addition to fetal tissues, including the brain [40,41].

In addition, Zika virus infection in pregnant women is also associated with adverse pregnancy outcomes including miscarriage, intrauterine growth restriction, perinatal death, and others [42,43]. As a result, Zika fever is now part of a group of tropical diseases that disproportionately affects maternal, fetal and reproductive health [38].

It has been found that Zika virus infection in pregnant women can cause a spectrum of congenital malformations that include: microcephaly (which determines incomplete brain development and reduced head size), severe global hypertonia, hyperexcitability, ocular changes, facial disproportion, and congenital contractures [34,36,44]. These congenital malformations are associated with disruption in fetal brain development during pregnancy, and may involve a disorder of neuronal and glial migration [4,39]. Our group conducted a study in which they observed histopathological changes, detection of the virus, and evidence of replication in all organs of the stillborn that were analyzed [45]. To date, studies that follow the development of children with congenital Zika syndrome are scarce and, therefore, the long-term effects of this condition remain unknown.

Another serious complication related to Zika virus infection in adults is the occurrence of Guillain-Barré syndrome—an autoimmune disease in which the immune system attacks part of the peripheral nervous system [46]. The pathogenesis of Guillain-Barré syndrome associated with Zika virus is not has been fully elucidated and may involve: direct neuropathogenic mechanisms, hyperacute immune response, and immune dysregulation [45,47,48].

## 4. The Placenta

After the epidemic and the state of emergency from 2015, there was an intense search and better understanding of the role of the placenta in ZIKV infection. Vertical transmission of ZIKV suggests tropism by cells of the placenta. The placenta is characterized as a temporary and chimeric organ, formed during gestation from maternal and fetal tissue, whose functions are essential for a healthy pregnancy. This organ is responsible for nutrition, gas exchange, removal of toxic wastes, in addition to providing endocrine and immunological support to the fetus, regulating the physiology of the mother and fetus throughout gestation and during delivery [49]. Problems in placental formation can cause gestational complications that result in fetal or maternal morbidity and even mortality, such as preeclampsia and fetal growth restriction [50,51]. The placenta has a very complex structure formed by maternal tissue and several types of trophoblastic cells derived from the embryo. The trophoblastic cells are epithelial cells specialized for various functions during gestation [49]. The human placenta is of the hemochorial type, in which maternal blood is in contact with trophoblastic cells of fetal origin.

Human hemo-monochorionic placentation is characterized by a unique fetal invasion of trophoblasts into the decidua. The fetal portion of the placenta consists of trophoblasts and extraembryonic mesoderm (chorionic plate) that initially proliferate rapidly than the embryo after implantation. The functional structure of the placenta consists of the chorionic villi and has three layers: the layer of syncytiotrophoblasts (multinuclear), which lies on the surface; the layer of cytotrophoblastic cells (mononuclear) and the mesoderm with the endothelium of the fetal vessels (which separates maternal and fetal blood). The basal decidua is the maternal component of the placenta, formed from the endometrium. The vessels in this portion of the endometrium supply the intervillar spaces with blood. By the fourth month the junctional zone is established, in which the trophoblast (syncytial giant cells) and decidual cells mix, and is also rich in amorphous material. After this period, most of the trophoblastic cells degenerate, leaving only the decidual and chorionic plate, with the intervillous spaces. During the fourth and fifth months the decidua forms several decidual septa, which project into the intervillous spaces [52,53]. Thus, maternal and fetal blood do not mix, except for the rupture of capillary walls, which happens rarely outside the delivery situation. This separation between fetal and maternal blood is called the placental membrane or barrier. It is guaranteed by four layers: fetal endothelium, the connective tissue in the villous axis with mesenchymal cells and fibroblasts, the cytotrophoblast and the syncytiotrophoblast. However, as the placenta matures, this layer becomes increasingly thinner to facilitate the exchange of products across the placental membrane, but compact enough to prevent many types of infections [52,53,54].

## 5. Placental Immune Cells

The immune status of the pregnant woman was long understood to be suppressed, but current works show evidence that immune responses at the maternal–fetal interface are not simply suppressed, but are highly dynamic [55]. Pregnancy is a major immunological paradox, given that the fetus and placenta consist of a semi-allogeneic graft, which should be automatically rejected by an immunologically competent host. However, the fetus is protected against immunological aggression, suggesting that there are complex adaptations on both sides so that the immune system acts based on tolerance rather than rejection [56]. The human placenta is characterized by the establishment of immunoprivileged fetal-maternal interfaces, where fetal tissues are in close contact with the maternal immune system [52]. During pregnancy, the decidua basalis contains a rich immunological infiltrate, which affects and can be affected according to the dynamics of the maternal–fetal interface. The two main interfaces are the decidua itself which serves as an anchor point for the placenta and the intervillous space [52,56].

It has already been noted that a pro-inflammatory environment is necessary even before implantation, rather than as the result of successful implantation. Deciduous leukocytes, especially dendritic and natural killer (NK) cells, assist in the apposition, adhesion, and invasion of the blastocyst [56,57,58]. As the blastocyst invades the endometrium and the decidua is formed, trophoblastic cells increase the population of decidual leukocytes by chemotaxis, via secretion of cytokines such as CXCL12, CXCL8, TGF-β and CCL2, and induces the differentiation of NK cells, macrophages and CD4+ T cells through the secretion of IL-15, GM-CSF and TGF-β, respectively [55,59,60].

Approximately 70% of decidual leukocytes are natural killer (NK) cells, 20 to 25% are macrophages, 1.7% are dendritic cells, and approximately 3 to 10% are T cells and mast cells. B cells are found, but in low quantity [55]. Even with this pro-inflammatory feature, maternal tolerance to fetal tissue is established, in part, by the unique behavior of decidual dendritic cells, which do not migrate to the maternal lymphatic vessels after exposure to fetal antigen, contrary to classical antigen presentation by normal dendritic cells. What happens in pregnancy is a unique event, in which the fetal antigens arrive without being processed or bound to any cell in the lymph nodes through passive transport and are presented to T cells by lymph node resident dendritic cells, a paradigm that does not produce significant immune responses [61,62].

That is, the pro-inflammatory environment in early pregnancy supports trophoblastic invasion and decidual remodeling, a process necessary to create a placenta that can supply and support the fetus throughout pregnancy. In the second trimester, placental growth slows down, and the peripheral environment changes and becomes anti-inflammatory. This allows maximum oxygen transfer to the fetus and corresponds to the period of substantial fetal growth. Hofbauer cells (placental resident macrophages) and regulatory T cells help modulate the environment by secreting anti-inflammatory cytokines and preventing the effector type immune response directed toward fetal tissue [60,63].

The innate immunity response is the first line of host immune defense against viral infections, with activation of immune cells and inflammatory mediators’ production [64]. As mentioned, immune cells that are part of the innate response present in the placenta have a role that still needs to be better investigated in ZIKV infection (Figure 1). We will discuss further what is known of the involvement of each of them below.

## 6. Placenta and Vertical Transmission

Vertical transmission is a major concern in ZIKV infection. Congenital infections have never been directly associated with flavivirus infections, and our understanding of congenital infections in the 2015–2017 Zika outbreak was based only on retrospective studies of other diseases. Due to the little knowledge regarding ZIKV infection in the placenta, studying this organ in infected pregnant women as well as in vitro models can provide important information. These may help to elucidate how transmission occurs, what the consequences are for the placenta and the fetus, because after all, as mentioned earlier, there is an epidemiological prospect of possible new outbreaks of this disease.

The Axl receptor, from the TAM receptor tyrosine kinase family, appears to be one of the most important in the entry of ZIKV into different cell types. Tabata et al. [65], conducted a study in explanted placental cells and cell lines, in which they observed that many of the placental cells (cytotrophoblasts, fibroblasts, decidual and endothelial cells), as well as the cells of the immune system that were present in the tissue—such as macrophages and dendritic cells—possessed the different surface receptors to which flaviviruses are able to bind and mediate their entry into the cell by endocytosis. These receptors include TAM family proteins (tyrosine kinases) such as Tyro3, Axl, and Mertk; TIM1, an immunoglobulin produced by T cells; and DC-SIGN proteins on fibroblasts, which allowed the different cells to be susceptible to ZIKV infection [65].

Some recent studies, most of them being carried out by our group, have identified the susceptibility and permissiveness of placenta cells, with the detection of the ZIKV genome and proteins in immune cells of the intervillous space, decidua and chorionic villus cells and in fetal tissues in stillbirths, including the liver, kidney, skin and brain [7,41,45,66,67,68,69]. Our group identified that the virus is persistent in placental tissue, as its replication in trophoblastic cells was identified months after the period of viremia [66,67]. Histological damage was found in the placentas such as villous immaturity, fibrin deposition, inflammatory infiltrate and areas of calcification. The increased cellularity (CD68+ cells and TCD8+ lymphocytes) detected in the tissue and the expression of local pro-inflammatory cytokines such as IFN-γ and TNF-α and other mediators, such as RANTES/CCL5, MMPs and VEGFR-2, support the characteristic placental inflammation and dysfunction caused by the virus [45,66]. Furthermore, we noted that decreases in the neurotrophin BDNF may modulate fetal neuronal damage, as its expression was lower in the placentas of patients who had babies with microcephaly [70].

## 7. Mastocytes and ZIKV Infection

Mast cells are abundant immune cells in the placenta and play an important role in general immunological reactions [71,72]. Because they are present in the skin and are always located close to blood vessels, mast cells may be among the first immune cells infected by ZIKV after the mosquito’s meal on the skin. The most common symptoms of zika include rash and itching, which are relieved by the administration of anti-allergic drugs (antihistamines), which leads us to believe that mast cells may play a role, as yet unclear, in the pathogenesis of the disease [73]. Because they are present in the endometrium and decidua, mast cells functionally contribute to implantation and its subsequent events, such as tissue remodeling and angiogenesis by secreting mediators and cytokines, after being activated by hormonal stimulation [74]. Still its location in the placenta, it could be one of the cells involved in vertical transmission in placental infection.

In a study performed in cell lineage (in vitro) and with placental tissue, our group observed for the first time that human mast cells are susceptible and permissive to infection and that ZIKV stimulates degranulation and release of cytokines and pro-inflammatory mediators [70]. Another in vitro study using a different mast cell lineage confirmed its permissiveness to ZIKV, as well as increased antibody-mediated infection in previous dengue infection [75]. This mast cell response may facilitate the installation of a pro-inflammatory environment in the places where these cells are found, including the placenta. Furthermore, the fact that they are permissive for virus replication in the human placenta suggests that this cell may contribute to vertical transmission. Ultrastructural aspects of infected cells in both tissues and cell cultures showed modifications in organelles, especially in the endoplasmic reticulum and mitochondria, and the presence of viral particles [66,67,70].

## 8. Dendritic Cells in ZIKV Placental Infection

Dendritic cells (DCs), which are believed to be initial sites of replication in mosquito-borne flaviviruses [76], are important components of the innate immune system and described as sentinels responsible for the antigen presentation, playing a major role in the interface of innate and adaptive immune response [77]. In the placenta, both mature (CD83+) and immature (DC-SIGN positive) dendritic cells are located in the basal and parietal decidua, supporting the implantation and functionally contributing to the homeostasis, angiogenesis and maintenance of the immunological tolerance during pregnancy [65,78,79].

Recent studies demonstrated that DCs are permissive to ZIKV infection through several adhesion factors such as DC-SIGN, AXL, TYRO3 [80,81,82,83], and could mediate an antagonistic response to type I interferon [84] and thus, they may play an important role in the pathogenesis of the disease and vertical transmission through their location in placental tissues. Tabata et al., studying the ZIKV replication in explants of anchoring villi and basal decidua from first-trimester human placentas, revealed the presence of E protein of Zika virus in dendritic cells present in the basal decidua, thus, confirming the infection of these cells, which could contribute to increased viral load [85]. Furthermore, these innate immune cells cannot migrate from the uterus and are thought to modulate NK and T cells by restricting cytotoxic/inflammatory responses in the placenta, which can lead us to think that dendritic cells infected by ZIKV could have a poor impact on placental development since its unique behavior in maternal–fetal tolerance, angiogenesis and implantation events [67,68,72,86,87,88].

In a study conducted with Rhesus macaques infected with ZIKV during different periods across pregnancy, full-term placentas isolated from the animals showed increased activated populations of innate immune cells in decidua, including dendritic cells, associated with an increase in the levels of inflammatory cytokines and mediators, suggesting the maintenance of a pro-inflammatory environment in response to viral infection [89,90]. Nevertheless, the exact function of DCs in these cases is yet to be determined. Moreover, studies regarding the exact role of innate immune cells, especially dendritic cells, in the placental tissues of ZIKV infections need to be further investigated.

## 9. The Importance of Macrophages and Hofbauer Cells in ZIKV Placental Infection

Hofbauer cells (HC) are placental-resident macrophages of fetal origin, which make up the maternal–fetal interface and the placental immune system. They are present in the stroma and adjacent to the trophoblast and fetal capillaries, a critical site for fetal protection against the vertical transmission of pathological agents [91,92]. These cells, together with the decidual macrophages, participate in placental homeostasis, promoting immune tolerance to the fetus [93] and participating in the immune response [93]. Furthermore, it is believed that HC participate in the development of villi, vasculogenesis, angiogenesis and stromal maturation [94,95].

Both Hofbauer cells and decidual macrophages resemble M2 macrophages, and therefore exhibit the anti-inflammatory profile that secretes mainly IL-10 and TGF-β, which helps in fetal immunotolerance [96,97,98,99,100,101,102,103,104,105]. On the other hand, it has been shown that when exposed to an inflammatory stimulus or to some infectious agent, Hofbauer cells can change the expression of molecules and exhibit an inflammatory profile [105]. This plasticity is possible because, despite exhibiting the anti-inflammatory profile, Hofbauer cells express high levels of TLRs (the inflammatory response is induced mainly through TRL-3 and TLR-4 and coactivators), in addition to being able to respond to infectious TLR agonists by producing cytokines and inflammatory mediators [106].

However, some studies have associated the secretion of pro-inflammatory cytokines/mediators by these cells with damage to the placental barrier and the presence of a fibrotic response associated with chronic inflammation, suggesting that placental macrophages play a role in placental/fetal inflammation [107].

These cells have been shown to be targeted and susceptible to infection by ZIKV, and other viral agents including cytomegalovirus, herpex simplex virus, chikungunya, SARS-CoV-2 [96]. They are even associated with the pathogenesis of several TORCH agents [101,108].

Several studies have identified that Zika virus infection leads to cell proliferation and Hofbauer cell hyperplasia [109] In addition, some studies have shown that infection-induced HBC hyperplasia coincides with the induction of type I interferon, pro-inflammatory cytokines [39], and an increase in TNF, which is also associated with a decrease in the suppressor molecule of the cytokine signaling SOCS1, in cases of chronic villitis caused by Zika virus infection [105].

It is believed that due to their basal immunoregulatory function, tissue location and susceptibility to infections, Hofbauer cells are important agents in the vertical transmission of the Zika virus and other pathogens [39,110]; however, the mechanism by which this event occurs remains unknown.

Therefore, it is possible that Hofbauer cells present an inadequate antiviral response to zika virus infection that might contribute to the worsening of pregnancy complications. However, further studies are needed to better understand the role of Hofbauer cells during viral infection.

## 10. Natural Killer Cells in ZIKV Placental Infection

The maternal–fetal interface is a complex environment essential for fetal development, it consists of maternal tissue and cells of fetal origin [111]. The maternal (deciduous) portion is rich in immune cells, including antigen-presenting cells, deciduous Natural Killer (dNK) [111,112]. Some studies have indicated that 70% of decidual immune cells during the first trimester are dNK [113,114,115]. dNK cells are a specialized type of Natural Killer cells and exhibit specific phenotypic and functional properties when compared to peripheral Natural Killer (pNK) cells [116,117]. During pregnancy, the number of dNK cells decreases, however, these cells remain present in the decidua of the placenta at term [117].

In the healthy placenta, dNK cells play an essential role in trophoblastic invasion and placental remodeling, through the secretion of chemokines (such as IL-8 and IP-10) and vascular modulation factors (such as VEGF-C, Arg1, Arg2 and TGF-β1) [118]. In addition, dNK cells participate in the immune response to placental infections by conserving trophoblastic cells [119,120,121,122].

Several recent studies point out the importance of dNK cells in the face of a viral or bacterial infection [56,57,122,123,124,125,126,127]. In a scenario of placental viral infection, it is believed that these cells change their secretory profile and become more cytotoxic and that this change is due to a direct cellular activation via the interaction of HLA molecules [57].

However, there are few studies that have analyzed the interaction of the zika virus with dNK cells. Santara et al. demonstrated that dNK cells were able to kill placental trophoblasts infected with the zika virus from the negative regulation of HLA-C/G inhibitory receptor ligands [128]. On the other hand, Glasner et al. observed that although ZIKV infection led to MHC I activation, somehow it was not able to activate dNK cells [129]. Another study also identified that in nonhuman primates, ZIKV infection caused a decrease in the number of dNK cells, suggesting that the p virus leads to local immunosuppression [111]. Therefore, the participation of dNK cells during placental infection by Zika virus remains unknown, and further studies are needed to analyze this interaction.

## 11. IFN-I Response in ZIKV Placental Infection

Type I interferons (IFN-I) are glycoproteins belonging to the class of cytokines and constitute the main innate antiviral defense of the body [115,130,131,132]. In general, IFN-I is constitutively expressed in lower concentrations in some tissues of the body so that at the time of a viral infection, concentrations rise rapidly [133]. IFN-I is known to be a pro-inflammatory cytokine. Different IFN-I subtypes, including IFN beta and IFN alpha, interact with the IFNAR complex (comprising the chains: IFNAR1 and IFNAR 2) to trigger a JAK-STAT-mediated signaling cascade that culminates in the transcription of interferon-stimulated genes (ISG) [134]. This IFNAR complex is expressed in almost all nucleated cells, allowing these molecules to trigger a systemic response, blocking viral replication [130,135,136,137].

The products of ISG restrict viral replication by several mechanisms that include the inhibition of viral entry into the cell, the inhibition of protein synthesis, the degradation of proteins or viral genetic material, alterations in cellular metabolism and others [136,138] In addition, some specific ISGs may play regulatory and immunomodulatory roles by encoding pattern recognition receptors (PRR) that detect viral molecules and modulate signaling pathways [139], encoding pro-apoptotic, anti-angiogenic molecules and participating in cell differentiation [132,136,140]. Moreover, to restrict viral replication, IFN-I performs extensive biological functions that have the potential to contribute to the pathogenesis of the disease [139,141,142].

In the placenta, IFN-I is an important immune modulator regulating inflammation, protecting against viral infections and contributing to fetal immunity [125,143]. These cytokines are involved in healthy pregnancies and are also important in the defense against pathogens. Any alteration in its expression can lead to an inefficient and uncoordinated immune system and consequently to pregnancy complications and congenital abnormalities [133]. Some studies have pointed out the importance of IFN-I as a protective factor at the maternal–fetal interface in situations of viral infection, directly impacting viral replication in the placenta as well as vertical transmission, especially of flaviviruses such as zika virus [144]. In immunocompetent mice, the response via IFN-I is essential for preventing infection, and the loss of signaling via IFN beta in the placenta can lead to exacerbated viral replication, fetal infection, hypersensitivity to bacterial products and even maternal mortality [128,143,145,146,147,148].

Monocytes/macrophages are critical cells for placental development. In ZIKV-infected placentas of pregnant women, the virus was detected in macrophages that were producing type I interferon [145]. In ZIKV-infected rhesus macaques, intermediate monocytes (CD14+ and CD16+) were found in high proportion in the maternal decidua and placental villous cells [146]. In addition, these cells seem to interfere in I IFN stimulated ISGs upregulation, such as IFI27, which was expressed in higher levels in the decidua and mainly in villous of ZIKV-infected rhesus macaques [146]. In vitro studies, at mid-pregnancy, chorionic villi also expressed more I IFNs compared with decidual tissues in the same period, as well as in chorionic villi at early gestation [147]. This observation suggests that as pregnancy progresses, there is a reduction in the susceptibility of chorionic villus to ZIKV, since there is an increase in the expression of innate antiviral factors in these tissues [147].

In humans, ZIKV NS5 inhibits type I IFN signaling by targeting the IFN-regulated transcriptional activator STAT2, resulting in proteasomal degradation [148]. However, in mice, STAT2 is resistant to ZIKV NS5 binding and consequent proteasomal degradation, which makes possible the clearance of ZIKV and avoiding viral spread to the developing fetus. STAT2 signaling appears to be relevant in the response to ZIKV, as STAT2-deficient mice (lacking I and III IFN signaling) are also susceptible to infections [149], as well as mice that are deficient in I IFN receptors (Figure 2) [150].

In mice lacking I IFN (*Ifnar1*−/−) signaling, the placenta exhibited severe damage, such as irregular shape, reduced fetal capillaries, presence of apoptotic trophoblasts and microvasculature destruction, leading to severe intrauterine growth restriction, ischemia and fetal demise [151]. Administration with anti-IFNAR blocking antibody in pregnant mice caused infection of the developing fetus, albeit to a lesser degree of severity without fetal demise [151].

However, IFN-I has pro-inflammatory properties that can be harmful during a chronic infection [148,152]. Several studies indicate that in pregnant women with a viral infection, the hyperstimulation of IFN-I can negatively impact the formation of the placenta, damaging the structure and function of the organ. In such scenarios, foci of apoptosis have been reported in the placental labyrinth region [141] damaged endothelium and placental barrier [141] as well as impaired proliferative capacity of trophoblastic stem cells [141] and similarly cause adverse conditions in the developing fetus including fetal growth restriction and intrauterine death [141,143,152,153,154].

Above all, IFN-I has been identified as one of the main agents of adverse effects in pregnancy, being closely related to fetal death [35,153]. R. Kwon et al. identified that a deregulation in signaling via IFN beta and that high concentrations of this cytokine resulted in intrauterine death. In the case of the alpha subtype (IFN alpha), there is only one study, using porcine models, that analyzes the relationship of this cytokine with gestational disorders [155]. In that study, the researchers found that although the offspring had no congenital lesions or malformations, they had excessive concentrations of IFN alpha, which had previously been associated with severe viral infection and immune dysfunction, as well as more than 100 genes with dysregulated transcription [144].

In addition, some studies suggest that uterine inflammation caused by subclinical TORCH infections [144,155] may trigger this type I IFN-mediated exacerbated response. [144,153]. Another factor that supports this trend is that pregnant women with interferonopathies (a class of diseases that result from genetic disorders and that lead to excessive production of IFN-I even in the absence of infection) often have a number of adverse conditions during pregnancy, including miscarriage, preterm birth, and increased risk of preeclampsia [141].

This IFN-I-mediated exacerbated response profile has immense potential to be detrimental to both fetal development and placental integrity [133,141,143,152,153,154]. However, it is important to emphasize that the IFN-I class plays an important role in antiviral defense [135]. Therefore, it may be that a balanced production of type I IFN is effective in fighting viral infection [153].

More studies are needed to help the scientific community in understanding the IFN response in front of a ZIKV infection, especially in pregnancy, in order to prevent adverse outcomes in futures outbreaks.

## 12. Conclusions

Understanding the host–pathogen relationships in ZIKV infections can be a challenging task. To control the infection in an effective way, orchestrating an early correct innate immune response is essential. Maternal–fetal transmission is a well-established event in the literature, and several innate immune cells present in the placenta are permissive of ZIKV infection. Therefore, it is possible that Hofbauer cells, decidual macrophages, mastocytes and dendritic cells act as sites of viral replication contributing to higher viral loads and transplacental infection. Nevertheless, ZIKV is capable of inducing a pro-inflammatory environment in placental tissue, activating innate immune cells and, thus, overlapping the antiviral response. Further studies are needed to clarify the mechanisms of immunopathogenesis of Zika virus in the placenta of pregnant women and their relationships with poor neonatal outcomes.

## Figures and Tables

**Figure 1 pathogens-11-01410-f001:**
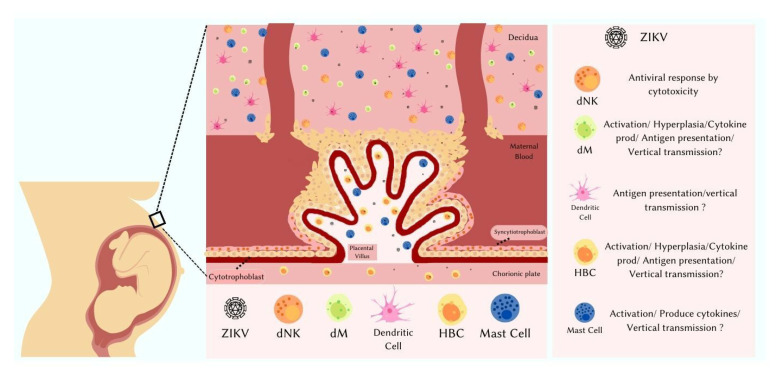
Schematic representation of the human maternal–fetal interface during Zika virus infection. Basal decidua with immune cells: decidual Natural Killer (dNK), maternal macrophages (dM), dendritic cells and mast cells. Chorionic villi contain trophoblast cells, mast cells, Hofbauer cells (HBC), and fetal capillaries surrounded by a layer of multinucleated syncytiotrophoblast cells. The chorionic villus is floating in the intervillous space, bathed in maternal blood. Maternal macrophages, Hofbauer cells, dendritic cells and mast cells are permissive and susceptible to ZIKV infection. Maternal macrophages and Hofbauer cells, when infected, are activated, go through hyperplasia, produce cytokines and participate in antigen presentation, and may be involved in vertical transmission. Dendritic cells participate in the immune response by presenting antigens and may be involved in vertical transmission. Mast cells, when infected, are activated, produce cytokines and may be involved in vertical transmission. Decidual Natural Killer cells produce an antiviral response by cytotoxicity.

**Figure 2 pathogens-11-01410-f002:**
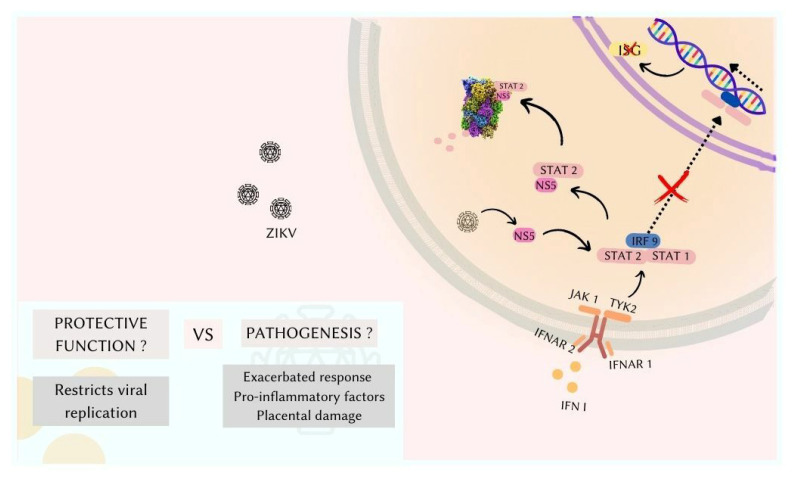
Schematic representation of the interaction between the Zika virus and the IFN-I response in humans. In healthy cells, via paracrine signaling INF-I binds to IFNAR and triggers the JAK-STAT cascade that induces the transcription of ISGs. Zika virus (ZIKV) infection inhibits the transcription of ISGs by IFN-I, as the non-structural protein NS5 of ZIKV binds to STAT2 and directs its degradation in the proteasome. IFN-I has an important antiviral function due to the potential to inhibit viral replication. However, against ZIKV infection, IFN-I was associated with exacerbated response by pro-inflammatory factors and associated with placental damage.

## Data Availability

Not applicable.

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
