# Peer review of "The Innate Defense in the Zika-Infected Placenta"

_pathogens, 2022, doi:10.3390/pathogens11121410_

Round 1

Reviewer 1 Report

This is a very interesting review article about Zika Virus infected placenta and innate defense mechanisms.

In this review the authors discussed about the some of the most recent findings about innate immune reactions to ZIKV, particularly in the placenta. The authors summarizes that The Zika virus (ZIKV), an arthropod-borne virus of the genus Flavivirus, was originally discovered in the serum of a sentinel Rhesus monkey in 1947 in Uganda, Africa.Since its discovery, the virus has caused significant epidemics in a number of nations has been associated with serious difficulties for expectant mothers and neonatal birth abnormalities, including the Congenital Zika Syndrome. All trimesters of pregnancy can experience maternal-fetal transmission of ZIKV, and the placenta's and its cells' functions in these situations are still poorly understood.

The paper is well organized and written, however, there are few suggestions:

Minor concerns:

1.      Line no 93, As with other arboviruses…. The authors should include a few names of arboviruses.

2.       Authors should increase the resolution of figures for better understanding.

Author Response

Thank you for your considerations about the manuscript. We highly appreciated the comments and opinion. We added the examples of arbovirus in  line 93, highlighted.

The images were in low resolution as they were added to the PDF. We have already uploaded the final versions in 300dpi and TIFF, ensuring better resolution.

Reviewer 2 Report

The manuscript titled "The innate defense in the zika infected placenta" by Arruda et al. is well-written and organized. The authors review the role of different immune cells involved in pregnancy and during Zika virus infection. However, the authors have missed including relevant citations (Line no. 396, PMID: 30905842).

Author Response

Thank you for your considerations about the manuscript. We highly appreciated the comments and opinion. We added the suggested reference in line 396, highlighted.